# Movement quantity and quality: How do they relate to pain and disability in dancers?

**Danica Hendry** [1]*, **Amity Campbell**[1], **Anne Smith**[1], **Luke Hopper**[2], **Leon Straker**[1], **Peter O'Sullivan**[1]

**1** School of Allied Health, Curtin University, Perth, Western Australia, **2** Western Australian Academy of Performing Arts, Edith Cowan University, Perth, Western Australia

☯ These authors contributed equally to this work.
* danica.hendry@curtin.edu.au

## Abstract

### Objective

This field-based study aimed to determine the association between pre-professional student dancers' movement quantity and quality with (i) pain severity and (ii) pain related disability.

### Methods

Pre-professional female ballet and contemporary dance students (n = 52) participated in 4 time points of data collection over a 12-week university semester. At each time point dancers provided self-reported pain outcomes (Numerical Rating Scale as a measure of pain severity and Patient Specific Functional Scale as a measure of pain related disability) and wore a wearable sensor system. This system combined wearable sensors with previously developed machine learning models capable of capturing movement quantity and quality outcomes. A series of linear mixed models were applied to determine if there was an association between dancers' movement quantity and quality over the 4 time points with pain severity and pain related disability.

### Results

Almost all dancers (n = 50) experienced pain, and half of the dancers experienced disabling pain (n = 26). Significant associations were evident for pain related disability and movement quantity and quality variables. Specifically, greater pain related disability was associated with more light activity, fewer leg lifts to the front, a shorter average duration of leg lifts to the front and fewer total leg lifts. Greater pain related disability was also associated with higher thigh elevation angles to the side. There was no evidence for associations between movement quantity and quality variables and pain severity.

### Discussion

Despite a high prevalence of musculoskeletal pain, dancers' levels of pain severity and disability were generally low. Between-person level associations were identified between dancers' movement quantity and quality, and pain related disability. These findings may reflect dancers' adaptations to pain related disability, while they continue to dance. This proof-of-

**Data Availability Statement:** Data cannot be shared publicly due to ethical considerations. All data is available for verification or other research following approval that such use complies with the

consent provided by participants. Researchers wishing to access the data can seek approval from the Curtin University Human Ethics Committee quoting project approval number HREC2017-0726. The contact for the Curtin University Human Ethics Committee is +61 8 9266 9223 or ethics@curtin.edu.au.

**Funding:** The first author is the recipient of an Australian Research Training Program Scholarship (RTP) (no grant number) for her PhD research of which this study was part of. The funders had no role in study design, data collection and analysis, decision to publish or preparation of the manuscript.

**Competing interests:** The authors have declared that no competing interests exist.

concept research provides a compelling model for future work exploring dancers' pain using field-based, serial data collection.

## Introduction

Dancers frequently experience musculoskeletal pain, which can be disabling, resulting in the need to modify or cease normal training [1]. Dancers are reported to undertake substantial workloads and perceive large workloads and related fatigue as leading causes of injury [2, 3]. In recent years, substantial attention has been placed on quantifying athlete training to assist in understanding the development and experience of pain and disability [4–7]. While athlete monitoring systems are commonly applied in many elite sports, it's only recently emerging within the field of dance, and only assesses *quantity* of dancers' movement [2, 3, 8–10]. One recent study has demonstrated that week to week increases in professional ballet dancers' movement quantity is was associated with the rate of overuse, time loss injury [9]. However, amongst pre-professional dancers, the relationship is less clear. While one study has observed that weekly reported injuries mirror fluctuations in dancers self-reported hours of weekly training, another has found no association [10, 11]. The lack of consensus may reflect how movement quantity is measured.

Previous research exploring dancers' movement quantity has focussed on quantifying cumulative workload from activity diaries and schedules, for example daily hours of training [9–13]. However it is recognised that these measures do not capture the movements that dancers perform within their training [9]. More recently, wearable sensors have been used to objectively determine the exercise intensity of dancers during their daily training [3, 8]. This work has demonstrated that while dancers participate in several hours of training per day, the majority of this time is spent at low to medium intensity exercise [3]. Both approaches offer useful insights, however, to date no method exists that provides detailed cumulative workload information such as the number of repetitions of movements that may be provocative of pain, for example the number of jumps or leg lifts performed.

Previous laboratory-based work has also demonstrated that the *quality* of movement may also be associated with pain and disability [14–16]. Movement quality refers to the specific biomechanical characteristics of movement and could include aspects such as forces, accelerations, range of movement and variability [14–16]. Specifically in dance movement quality, ground reaction force (GRF) during jumps, and thigh elevation angles and lumbar spine sagittal angles during leg lifting tasks, may be an important consideration for pain and disability [14–16]. Cross-sectional laboratory studies have shown that during jumping activities dancers achieve peak GRF 2–7 times body weight (BW) [17–21]. These substantial forces have been associated with the presence of lower limb pain [14]. Additionally, the large ranges of motion associated with leg lifting tasks have been suggested as contributing to the development of lower back and hip pain [22–24]. While considered gold standard, laboratory methods have low ecological validity, thus are more appropriate for once off screening tests as opposed to regular or ongoing monitoring. Regular monitoring of dancers' movement quality may assist in understanding the role of biomechanical demands in dancers' pain.

Recent developments in wearable sensor technology combined with the application of machine learning have allowed for the development of a dance-specific wearable sensor system capable of field-based measurement of movement quantity and quality [21, 25, 26]. This system enables field-based studies exploring the relationships of dancers' pain and disability with

movement quantity and quality within their naturalistic environment. Large longitudinal studies incorporating ongoing monitoring would enhance understanding of temporal association of changes in movement quality and quantity with musculoskeletal pain, which could be bidirectional. However, to justify larger studies it is important to understand if there are associations between movement quantity and quality within a dancer's normal training when they are experiencing pain, and if the system is capable of detecting these. Thus, this study aimed to estimate the association between pre-professional student dancers' movement quantity and movement quality with (i) pain severity, and (ii) pain related disability over the course of 1 university semester.

## Methods

### Study design

This was a field-based study in which repeated wearable sensor-based measures of movement quantity and quality, along with self-reported measures of pain and disability were collected at 4 time points across a 12-week university semester, in the lead up to and following a performance season. This research was approved by the institutional human research ethics committee (HREC2017-0726).

### Participants

All female dance students enrolled in the full-time dance courses at an Australian dance training institute (n = 100) were invited to participate in this study. The dancers were provided with an information session about the research and participant information sheets prior to 52 providing consent. Only female students were included in the study, as female and males demonstrate different pain and movement profiles [1, 27]. To be included in the study, dancers were required to be a minimum of 16 years old and enrolled in one of the university's full time dance training programs. These programs include extensive training in ballet and contemporary dance. All dancers provided written, informed consent prior to participation.

### Data collection

Prior to commencing training for the semester dancers had a brief (1–2 minute) interview with 1 of the researchers, either a qualified physiotherapist or a final year physiotherapy student, both with backgrounds in dance. Demographic and anthropometric information collected by interview included year of training enrolment (first, second or third), age they commenced dancing, dance stream (ballet or contemporary) and height and weight.

Dancers participated in 4 separate days (time points) of data collection. Only 4 days of data collection were scheduled across the 12-week semester period (see Fig 1) to minimise dancer burden. Data collected on 10–12 dancers each day, on a day with scheduled ballet technique class.

On each time point of data collection, dancers independently completed a short electronic survey detailing any current pain they were experiencing and were fitted with a previously developed wearable sensor system, capable of field-based movement quantification.

### Pain severity and pain related disability

Using the Self Estimated Functional Inability because of Pain Screening questionnaire (SEFIP) [28], dancers reported the anatomical location(s) of their pain in Qualtrics (Qualtrics, Seattle, WA, USA). Dancers were requested to report any pain, irrespective of whether it affected their ability to dance. If the dancer reported multiple locations, they were asked to identify the body

| | Normal Classes | | | | | | Rehearsal/ Production Period | | Performance Period | | Normal Classes | |
|---|---|---|---|---|---|---|---|---|---|---|---|---|
| Week | 1 | 2 | 3 | 4 | 5 | 6 | 7 | 8 | 9 | 10 | 11 | 12 |
| | Data Collection 1 | | | Data Collection 2 | | | | Data Collection 3 | | | | Data Collection 4 |
| Interview | X | | | | | | | | | | | |
| Pain Measures | X | | | X | | | | X | | | | X |
| Movement Quantity | X | | | X | | | | X | | | | X |
| Movement Quality | X | | | X | | | | X | | | | X |

**Fig 1. Data collection time periods across a university semester.**

region which was bothering them the most. This was considered their most bothersome pain and self-report of pain intensity and pain related disability for that time point was made with reference to this pain location. For each dancer, the location of most bothersome pain could differ over the 4 time points.

For their most bothersome pain dancers were asked to rate the intensity of their pain using a Numerical Rating Scale (NRS) (0–10 scale) [29]. NRS scores reported at each time point were used as a continuous variable indicating pain severity for aim 1, where higher scores indicated greater pain severity. The NRS has been determined as a reliable and valid measure of musculoskeletal pain [29].

Dancers completed the Patient Specific Functional Scale (PSFS) [30, 31], whereby they identified up to 3 self-selected important activities that they are unable to do or are having difficulty with as a result of their most bothersome pain. They then scored each activity from 0 to 10, where 0 indicated they were unable to perform the activity at all and 10 indicated that they were able to perform the activity at the same level as before the problem. PSFS scores reported at each time point were used as a continuous variable indicating pain related disability for aim 2. Lower scores indicated greater disability. For each dancer, the nominated activities could differ across the 4 time points. The PSFS has been determined as a reliable and valid measure of musculoskeletal disability [31].

Additionally, the presence of pain related disability at any of the time points was used to describe the sample. Pain was considered disabling for scores of less than 7 in the PSFS for at least 1 activity.

## Movement quantity and quality

Dancers were fitted with a wearable sensor system consisting of 6 Actigraph GT9x Link (Actigraph, Pensocola, FL) wearable inertial measurement units, which include an accelerometer, gyroscope and magnetometer, operated at 100Hz, at previously detailed anatomical landmarks (thoracic spine, sacrum, bilateral thigh and bilateral shin) in order to estimate movement quantity and quality [21, 25]. Dancer's movement quantity was defined as the number of movements that a dancer performed or the time spent performing the movements, and movement quality was defined as the biomechanical characteristics of the movement. Specifically, the previously developed and validated system utilised machine learning models applied to raw data to estimate every occurrence of jumping (unilateral and bilateral jumps) and leg

lifting (to the front, side and back), as measures of specific movement quantity [25]. It then output peak GRF during jumps (with a potential error of 0.24BW for unilateral landings and 0.21BW for bilateral landings, as well as thigh elevation angles and lumbar spine sagittal angles during leg lift tasks (with a potential error of 6.8˚ and 5.7˚ respectively during leg lifts), as measures of movement quality [21, 26]. In addition, the accelerometer data collected with the sacrum sensor was used to determine the physical activity intensity completed by the dancers using established physical activity intensity cut points [32], this was utilised as a measure of general movement quantity.

Following data collection, the sensor data was processed and cleaned as described in the flow diagram in Fig 2. Extensive and comprehensive data cleaning was applied to remove any movements that were misclassified. Specific parameters applied for data cleaning are detailed in Fig 2. Initially, it was proposed that the dancers' quantity and quality of movement would be analysed over a full day of training. However due to the complex computational time we focussed on the best estimate of the dancer's load, which was their ballet class. Quantity and quality of movement within a single 1.5-hour ballet class at each time point was analysed. General and specific quantity of movement variables were considered (Fig 2).

### Statistical analyses

Sample demographics were summarised with descriptive statistics.

For aim 1, a series of linear mixed models were used to estimate the association between quantity and quality of movement and pain severity. Pain severity (NRS) was used as the dependent variable and, in separate models, quantity and quality of movement variables were used as the independent variable.

The level 1 unit of observation was occasion (4 measures over the semester), nested in participants as the level 2 unit of observation. For each model, within-person and between-person level associations were estimated separately using subject mean centering [33]. Between-person analysis seeks to explain how much the variability between the pain scores of different people is a function of differences in levels of movement between those people, whereas within-person analysis seeks to explain how much of the variability in pain a single person over time is a function of that person's levels of movement over time.

A likelihood ratio test was conducted to assess support for a random slope over a random intercept model, and nonlinear and linear effects for time were also evaluated. The association of year level (first, second or third year) and stream of dance (ballet or contemporary) with pain severity was assessed to evaluate the potential confounding of these variables on the between-person associations between pain severity and quantity and quality of movement variables. Regression coefficients with accompanying 95% confidence intervals and p-values are reported.

The same analyses were conducted for aim 2, using pain related disability (PSFS) as the dependent variable. All analyses were conducted using Stata/IC 16.0 for Windows (StataCorp LLC; College Station TX USA).

### Results

Of the 52 dancers who consented to participate in this study, two dancers withdrew from the dance program after the second data collection period, and two more elected not to wear sensors for the final data collection due to skin irritation. These dancers were still included in the analysis, whereby the analysis model accounted for missing data. One dancer's data at all 4 time points was not usable as the human activity recognition machine learning model could not provide an output, and thus was excluded from analysis.

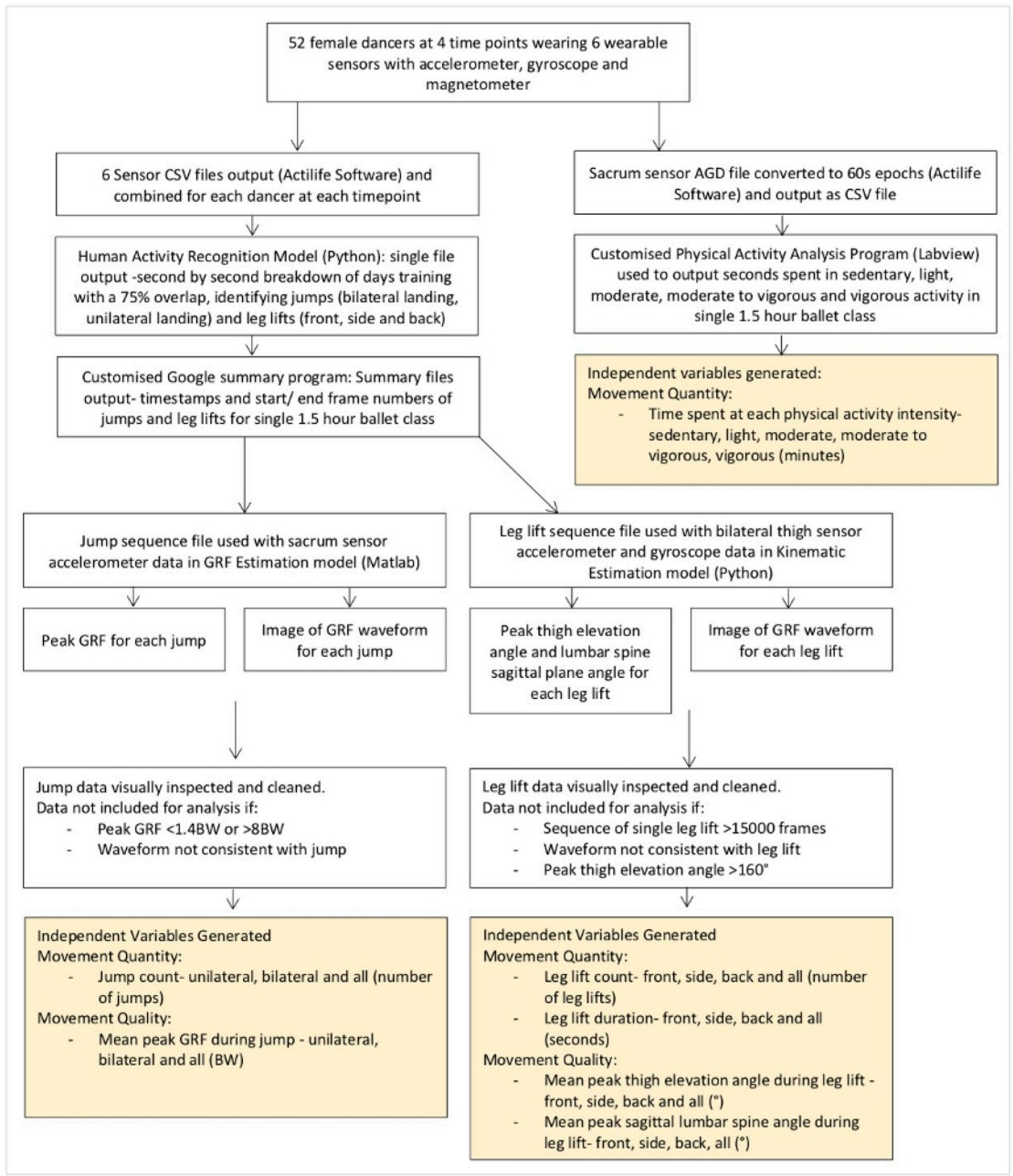

**Fig 2. Flow diagram representing sensor data processing steps and variables generated.**

### Participant characteristics

Participant characteristics are displayed in Table 1. There were similar numbers of ballet and contemporary focussed dancers represented in the sample, however there were more first-year dancers than second- and third-year dancers.

Throughout the study, 50 of the 52 dancers experienced pain, and 26 of these reported PSFS scores of less than 7/10, classified as disabling pain. The frequency of reporting of all pain sites, most bothersome pain sites and disabling pain sites across all 4 time periods is

**Table 1. Participant characteristics.**

| Characteristic | Mean (SD) |
|---|---|
| Age (years) | 18.4 (1.1) |
| Height (cm) | 168.4 (5.4) |
| Weight (kg) | 59.5 (5.8) |
| Years of dance training (years) | 13.7 (3.0) |
| Year group (first/ second/ third) | 26/16/10 |
| Stream (ballet/ contemporary) (n) | 28/ 24 |

demonstrated in Fig 3. When considering all pain presentations, the lower back was most commonly affected, followed by the hip and the foot and ankle. The lower back was most commonly nominated as the most bothersome, followed by the foot and ankle and 22 dancers reported multiple pain sites as most bothersome. The foot and ankle were most common area of disabling pain (< 7 on PSFS), followed multiple pain sites (9 presentations) and then the lower back. Only 2 of the dancers with disabling pain completely stopped dancing due to their pain. Both were due to acute traumatic injuries, the only two across the course of the study.

## Relationship between movement quantity and quality with pain severity and pain related disability

The overall mean values for each variable at each time point are demonstrated in Table 2. Pain severity and pain related disability remained fairly constant over the 4 time points, and there was no statistical evidence for linear or non-linear effects for time for either outcome in linear mixed models (pain severity: coefficient -0.07, 95%CI: -0.32, 0.18, p = 0.58, pain related disability: coefficient -0.10, 95%CI: -0.41, 0.22, p = 0.55).

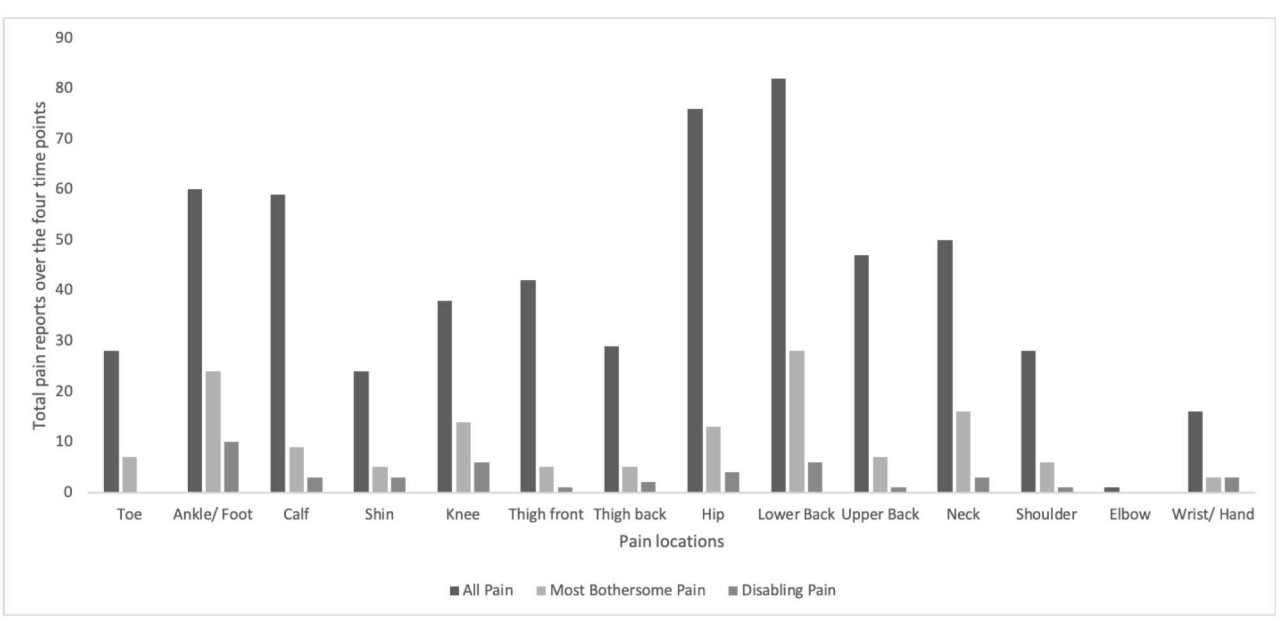

**Fig 3. Frequency of pain reports by pain location.**

**Table 2. Overall mean values for each variable at each time point.**

| | Mean (SD) score at each time point | | | |
|---|---|---|---|---|
| | 1 | 2 | 3 | 4 |
| **Pain** | | | | |
| Pain Related Disability (PSFS) | 8.2 (2.7) | 8.2 (3.0) | 7.8 (3.3) | 8.1 (2.9) |
| Pain Severity (NRS) | 3.5 (2.1) | 3.1 (2.1) | 3.2 (2.3) | 3.2 (2.3) |
| **Movement Quantity** | | | | |
| *General: Intensity* | | | | |
| Sedentary (mins) | 29.5 (15.6) | 29.7 (10.9) | 25.6 (10.1) | 21.8 (7.7) |
| Light (mins) | 62.2 (8.9) | 56.3 (7.2) | 57.5 (7.5) | 62.4 (11.0) |
| Moderate (mins) | 7.5 (4.7) | 11.8 (7.0) | 13.4 (7.0) | 9.6 (4.8) |
| Vigorous (mins) | 1.4 (1.3) | 2.4 (2.3) | 1.8 (1.2) | 2.9 (2.7) |
| Moderate to Vigorous (mins) | 8.9 (5.3) | 14.2 (8.4) | 15.2 (7.4) | 11.9 (6.3) |
| *Specific: Leg lift* | | | | |
| Duration front (secs) | 82.9 (64.5) | 118.0 (71.1) | 88.1 (56.3) | 115.8 (68.3) |
| Duration side (secs) | 28.3(29.1) | 47.2 (31.6) | 48.1 (30.7) | 45.4 (34.8) |
| Duration back (secs) | 36.6 (32.1) | 70.9 (48.9) | 82.1 (56.0) | 68.5 (44.5) |
| Duration all (secs) | 143.2 (96.7) | 236.0 (108.9) | 217.3 (106.3) | 228.2 (105.5) |
| Count front | 28.2 (20.8) | 35.6 (16.2) | 32.0 (19.0) | 40.1 (21.7) |
| Count side | 11.2 (9.0) | 18.0 (11.6) | 19.8 (11.0) | 18.5 (15.1) |
| Count back | 16.6 (15.9) | 28.9 (15.1) | 35.4 (23.2) | 28.1 (19.4) |
| Count all | 54.3 (35.9) | 82.5 (27.7) | 86.7 (39.6) | 86.1 (40.7) |
| *Specific: Jumps* | | | | |
| Count Unilateral | 20.1 (21.7) | 35.3 (27.9) | 41.2 (33.6) | 65.6 (67.9) |
| Count bilateral | 7.9 (12.5) | 19.6 (19.2) | 20.0 (22.8) | 28.1 (39.3) |
| Count all | 25.7 (26.1) | 53.4 (38.4) | 59.4 (46.5) | 90.7 (92.6) |
| **Movement Quality** | | | | |
| *Leg lifts* | | | | |
| Thigh elevation front (°) | 93.7 (19.1) | 91.9 (7.8) | 90.3 (11.1) | 84.4 (11.6) |
| Thigh elevation side (°) | 110.9 (19.8) | 104.6 (13.1) | 110.9 (17.5) | 102.6 (19.6) |
| Thigh elevation back (°) | 91.2 (23.1) | 84.2 (10.9) | 84.5 (13.1) | 80.3 (12.5) |
| Thigh elevation all (°) | 96.7 (19.7) | 91.6 (8.0) | 91.9 (10.5) | 86.8 (13.2) |
| Lumbar sagittal front (°) | -6.0 (4.34) | -6.7 (2.7) | -5.8 (3.4) | -4.7 (2.6) |
| Lumbar sagittal side (°) | -5.3 (4.9) | -4.9 (2.4) | -4.3 (2.5) | -3.8 (2.8) |
| Lumbar sagittal back (°) | 29.4 (4.6) | 29.9 (5.7) | 30.4 (4.2) | 31.7 (3.6) |
| *Jumps* | | | | |
| GRF Unilateral (BW) | 2.7 (0.4) | 2.6 (0.3) | 2.6 (0.3) | 2.6 (0.4) |
| GRF Bilateral (BW) | 2.7 (0.5) | 2.8 (0.6) | 2.6 (0.4) | 2.7 (0.4) |
| GRF All (BW) | 2.7 (0.4) | 2.7 (0.3) | 2.6 (0.3) | 2.6 (0.3) |

Secs: Seconds; Mins: Minutes; BW: Body weight; GRF: Ground reaction force

Of the large number of movement parameters examined, we identified only a small number of modest associations between dancers' movement quantity and quality and dancers' self-reported pain outcomes. In summary, there were no significant between-person level associations for pain severity, however increased pain related disability was associated with higher levels of light activity and a lower duration and count of leg lifts to the front and all leg lifts, and higher thigh elevation angles during side leg lifts. At a within-person level there were no significant findings for either pain severity or pain related disability. Results of the between-person

and within-person analysis for movement quantity and quality are demonstrated in table form in Table 3.

#Negatively signed coefficient indicated increase in the independent variable associated with increase in pain related disabilityWhen considering movement quantity, there was no evidence of between-person associations with pain severity after adjusting for year and stream.

When considering movement quantity, there was some evidence of very modest between-person associations with pain related disability on adjusted analysis. A 1-minute increase in light activity was associated with a reduction in patient specific functional scale of -0.15 points, 95%CI: -0.26, -0.03, p = 0.02) equating to an association with increased pain-related disability.

Additionally, a 10-count increase in front leg lifts was associated with a decrease in pain related disability of 0.66 points (95%CI: 0.13, 1.19 p = 0.02). Further, a 10-count increase in all leg lifts was associated with a decrease in pain related disability of 0.40 points (95%CI: 0.03, 0.76, p = 0.03).

When considering movement quality, there was evidence of a between-person association with pain related disability on adjusted analysis, with a 10˚ increase in thigh elevation angle during side leg lifts was associated with an increase in pain related disability of 0.83 points. (95%CI: -1.57, -.0.09, p = 0.03.

## Discussion

This field-based study utilised wearable sensor technology combined with machine learning methods to repeatedly monitor the movement quantity and quality for 52 dancers during their ballet classes over 4 time periods, in the lead up to and following a performance in a 12-week university semester. Some associations between self-reported pain outcomes with field-based movement quantity and quality were identified at the between-person level. While there was no evidence of associations with dancers' self-reported pain severity, a few modest associations were identified between some movement factors and pain related disability. There was no evidence of changes over the 4-time points time in either pain or pain related disability. The methods used in this study provides a platform for further longitudinal research using continuous dancer monitoring to understand the complexities related to the development of, and responses to pain and pain related disability in dancers.

There was a high prevalence of musculoskeletal pain reported within this sample of dancers, with almost all dancers (50 out of 52) reporting having musculoskeletal pain at some point during the semester. The foot and ankle, and lower back were the sites most commonly reported for presence of pain, most bothersome pain and disabling pain. These findings are consistent with previous literature, where a systematic review has demonstrated a 14–57% prevalence for foot and ankle pain and 62% for low back pain [34]. Further, the prevalence of disabling pain was lower, with half the dancers reporting disabling pain at some point during the semester. This is consistent with previous work, where the prevalence of dance related pain is influenced by how it is defined [35]. In our cohort, while half of the dancers experienced disabling pain across the semester, only two ceased dancing completely for at least 1 day due to their pain. The rest persisted, with some training modifications, which may reflect relatively low levels of pain and disability. Alternatively, it may reflect a culture of persisting in dance activities regardless of pain [36, 37], or that movement is only modestly associated with pain.

Interestingly, there was no evidence for associations between pain severity and both movement quantity and quality when year and stream were accounted for. This result suggests that irrespective of pain intensity, dancers continue to engage in a similar amount of training and with the same movement quality.

**Table 3. Results of linear mixed models examining associations between quantity and quality of movement with pain severity and pain related disability.**

| | | Pain Severity | | | | Pain Related Disability | | | |
| --- | --- | --- | --- | --- | --- | --- | --- | --- | --- |
| | | Greater scores indicate greater pain severity | | | | Lower scores indicate greater pain related disability | | | |
| | | Unadjusted Analysis | | Analysis adjusted for year and stream | | Unadjusted Analysis | | Analysis adjusted for year and stream | |
| | | Coefficient (95% Confidence Interval)* | P | Coefficient (95% Confidence Interval)* | P | Coefficient (95% Confidence Interval)# | P | Coefficient (95% Confidence Interval)# | P |
| **Movement Quantity: General** | | | | | | | | | |
| Sedentary (mins) | Between | -0.04 (-0.08, 0.01) | 0.07 | -0.01 (-0.07, 0.04) | 0.62 | -0.02 (-0.08, 0.04) | 0.46 | -0.04 (-0.11, 0.04) | 0.31 |
| | Within | 0.01 (-0.02, 0.04) | 0.60 | | | -0.01 (-0.06, 0.03) | 0.50 | | |
| Light (mins) | Between | 0.08 (-0.01, 0.17) | 0.06 | 0.06 (-0.03, 0.14) | 0.19 | **-0.16 (-0.27, -0.05)** | **0.01** | **-0.15 (-0.26, -0.03)** | **0.02** |
| | Within | -0.01 (-0.04, 0.02) | 0.43 | | | 0.01 (-0.03, 0.05) | 0.63 | | |
| Moderate (mins) | Between | 0.09 (-0.01, 0.18) | 0.08 | 0.04 (-0.10, 0.17) | 0.60 | 0.07 (-0.07, 0.20) | 0.32 | 0.12 (-0.07, 0.31) | 0.21 |
| | Within | 0.012 (-0.04, 0.07) | 0.65 | | | -0.01 (-0.78, 0.07) | 0.88 | | |
| Vigorous (mins) | Between | **0.32 (0.02, 0.63)** | **0.04** | 0.29 (-0.22, 0.79) | 0.27 | 0.15 (-0.28, 0.58) | 0.50 | 0.23 (-0.52, 0.98) | 0.55 |
| | Within | -0.09 (-0.25, 0.07) | 0.27 | | | 0.00 (-0.22, 0.21) | 0.98 | | |
| Moderate-Vigorous (mins) | Between | 0.08 (0.00, 0.15) | 0.05 | 0.05 (-0.07, 0.16) | 0.44 | 0.05 (-0.060, 0.158) | 0.38 | 0.09 (-0.08, 0.27) | 0.29 |
| | Within | 0.00 (-0.05, 0.05) | 0.96 | | | 0.00 (-0.068, 0.064) | 0.96 | | |
| **Movement Quantity: Leg Lifts** | | | | | | | | | |
| Duration front (secs)[a] | Between | 0.01 (-0.11, 0.12) | 0.94 | -0.01 (-0.11, 0.11) | 0.10 | **0.18 (0.02, 0.32)** | **0.03** | **0.19 (0.04, 0.34)** | **0.02** |
| | Within | -0.01 (-0.05, 0.03) | 0.67 | | | 0.001 (-0.04, 0.07) | 0.64 | | |
| Duration side (secs)[a] | Between | -0.11 (-0.12, 0.34) | 0.37 | -0.02 (-0.31, 0.27) | 0.91 | 0.02 (-0.31, 0.35) | 0.92 | 0.19 (-0.23, 0.63) | 0.37 |
| | Within | 0.06 (-0.15, 0.03) | 0.20 | | | 0.05 (-0.06, 0.17) | 0.35 | | |
| Duration back (secs)[a] | Between | -0.02 (-0.17, 0.19) | 0.87 | -0.01 (-1.66, 0.19) | 0.89 | 0.09 9 (-0.17, 0.36) | 0.48 | 0.09 (-0.17, 0.36) | 0.49 |
| | Within | -0.01 (-0.07, 0.05) | 0.70 | | | 0.02 (-0.05, 0.010) | 0.54 | | |
| Duration all (secs)[a] | Between | 0.02 (-0.05, 0.09) | 0.58 | 0.01 (-0.06, 0.08) | 0.82 | 0.08 (-0.02, 0.18) | 0.13 | 0.10 (-0.01, 0.21) | 0.06 |
| | Within | -0.01 (-0.03, 0.02) | 0.49 | | | 0.02 (-0.02, 0.05) | 0.37 | | |
| Count front[b] | Between | 0.14 (-0.21, 0.48) | 0.44 | -0.02 (-0.39, 0.34) | 0.90 | **0.51 (0.02, 0.99)** | **0.04** | **0.66 (0.13, 1.19)** | **0.02** |
| | Within | -0.03 (-0.19, 0.12) | 0.66 | | | 0.08 (-0.11, 0.28) | 0.39 | | |
| Count side[b] | Between | 0.29 (-0.31, 0.89) | 0.34 | -0.30 (-0.18, 0.57) | 0.50 | 0.13 (-0.73, 0.99) | 0.77 | 0.85 (-0.45, 2.15) | 0.12 |
| | Within | -0.14 (-0.38, 0.09) | 0.23 | | | 0.11 (-0.20, 0.42) | 0.49 | | |
| Count back[b] | Between | 0.24 (-0.15, 0.64) | 0.24 | 0.10 (-0.34, 0.53) | 0.67 | 0.16 (-0.42, 0.73) | 0.59 | 0.19 (-0.46, 0.84) | 0.56 |
| | Within | -0.02 (-0.16, 0.13) | 0.83 | | | 0.09 (-0.10, 0.27) | 0.36 | | |
| Count all[b] | Between | 0.13 (-0.06, 0.32) | 0.17 | 0.01 (-0.24, 0.26) | 0.93 | 0.18 (-0.09, 0.45) | 0.18 | **0.40 (0.03, 0.76)** | **0.03** |
| | Within | -0.02 (-0.10, 0.05) | 0.55 | | | 0.06 (-0.04, 0.16) | 0.23 | | |
| **Movement Quantity: Jumps** | | | | | | | | | |
| Count Jump unilateral[b] | Between | 0.11 (-0.05, 0.27) | 0.16 | 0.03 (-0.16, 0.23) | 0.73 | 0.02 (-0.23, 0.26) | 0.89 | 0.01 (-0.30, 0.32) | 0.93 |
| | Within | 0.01 (-0.07, 0.09) | 0.81 | | | 0.05 (-0.05, 0.14) | 0.34 | | |
| Count Jump bilateral[b] | Between | 0.13 (-0.20, 0.46) | 0.43 | 0.07 (-0.31, 0.45) | 0.71 | -0.11 (-0.63, 0.40) | 0.66 | -0.24 (-0.86, 0.37) | 0.44 |
| | Within | 0.12 (-0.01, 0.25) | 0.07 | | | 0.01 (-0.15, 0.16) | 0.94 | | |
| Count Jump all[b] | Between | 0.08 (-0.04, 0.20) | 0.19 | 0.03 (-0.12, 0.18) | 0.71 | -0.03 (-0.21, 0.16) | 0.78 | -0.06 (-0.30, 0.18) | 0.61 |
| | Within | 0.03 (-0.02, 0.09) | 0.25 | | | 0.03 (-0.04, 0.10) | 0.38 | | |
| **Movement Quality: Leg Lifts** | | | | | | | | | |
| Thigh elevation all (°)[c] | Between | 0.14 (-0.75, 0.47) | 0.64 | -0.05 (-0.79, 0.70) | 0.90 | -0.31 (-1.18, 0.56) | 0.48 | -0.08 (-1.21, 1.05) | 0.89 |
| | Within | -0.04 (-0.18, 0.25) | 0.74 | | | -0.12 (-0.39, 0.16) | 0.41 | | |
| Thigh elevation front (°)[c] | Between | -0.40 (-1.01, 0.20) | 0.19 | -0.47 (1.17, 0.22) | 0.18 | 0.05 (-0.94, 0.83) | 0.91 | 0.29 (-0.77, 1.36) | 0.59 |
| | Within | 0.03 (-0.18, 0.23) | 0.80 | | | -0.04 (-0.31, 0.23) | 0.75 | | |
| Thigh elevation side (°)[c] | Between | 0.25 (-0.27, 0.72) | 0.31 | 0.31 (-0.20, 0.82) | 0.23 | **-0.83 (-1.47, -0.19)** | **0.01** | **-0.83 (-1.57, -0.09)** | **0.03** |
| | Within | 0.05 (-0.11, 0.20) | 0.54 | | | -0.08 (-0.28, 0.12) | 0.45 | | |

*(Continued)*

**Table 3.** (Continued)

| | | Pain Severity | | | | Pain Related Disability | | | |
|---|---|---|---|---|---|---|---|---|---|
| | | Greater scores indicate greater pain severity | | | | Lower scores indicate greater pain related disability | | | |
| | | Unadjusted Analysis | | Analysis adjusted for year and stream | | Unadjusted Analysis | | Analysis adjusted for year and stream | |
| | | Coefficient (95% Confidence Interval)* | P | Coefficient (95% Confidence Interval)* | P | Coefficient (95% Confidence Interval)# | P | Coefficient (95% Confidence Interval)# | P |
| Thigh elevation back (°)a | Between | 0.18 (-0.29, 0.65) | 0.46 | 0.33 (-0.16, 0.82) | 0.18 | - -0.57 (-1.22, 0.08) | 0.09 | -0.56 (-1.30, 0.18) | 0.14 |
| | Within | 0.06 (-0.12, 0.24) | 0.51 | | | 0.10 (-0.33, 0.13) | 0.40 | | |
| Lumbar sagittal front (°)c | Between | 0.12 (-1.88, 2.13) | 0.91 | 0.41 (-1.58, 0.241) | 0.69 | -1.88 (-4.75, 0.98) | 0.19 | -2.66 (-5.61, 0.30) | 0.08 |
| | Within | 0.04 (-0.83, 0.92) | 0.92 | | | 0.40 (-0.74, 1.53) | 0.49 | | |
| Lumbar sagittal side (°)c | Between | 0.67 (-2.29, 3.64) | 0.66 | 0.48 (-2.57, 3.53) | 0.76 | -2.30 (6.60, 2.00) | 0.29 | -3.77 (-8.36, 0.81) | 0.10 |
| | Within | -0.23 (-1.06, 0.59) | 0.58 | | | -0.25 (-1.31, 0.82) | 0.65 | | |
| Lumbar sagittal back (°)c | Between | -1.16 (-3.00, 0.68) | 0.22 | -0.69 (-2.53, 1.15) | 0.46 | -0.41 (-3.07, 2.25) | 0.76 | -0.50 (-3.23, 2.32) | 0.75 |
| | Within | -0.18 (-0.79, 0.43) | 0.56 | | | -0.011 (-0.91, 0.69) | 0.79 | | |
| **Movement Quality: Jumps** | | | | | | | | | |
| GRF unilateral (BW) | Between | 1.48 (-0.44, 3.40) | 0.13 | 0.72 (-1.50, 2.94) | 0.52 | 1.12 (-1.56, 3.80) | 0.41 | 1.73 (-1.46, 4.93) | 0.29 |
| | Within | -1.01 (-2.10, 0.08) | 0.07 | | | -0.29 (-1.70,1.13) | 0.69 | | |
| GRF bilateral (BW) | Between | 0.61 (-0.79,2.01) | 0.39 | 0.49 (-0.95, 1.92) | 0.51 | -0.31 (-2.30, 1.67) | 0.76 | -0.65 (-2.75, 1.45) | 0.55 |
| | Within | 0.75 (-0.86, 2.35) | 0.36 | | | -0.23 (-1.10, 0.64) | 0.61 | | |
| GRF All (BW) | Between | 1.44 (-0.42, 3.31) | 0.13 | 1.10 (-0.84, 3.04) | 0.27 | 0.27 (-2.30, 2.84) | 0.84 | -0.01 (-2.76, 2.74) | 1.00 |
| | Within | -0.99 (-2.05, 0.07) | 0.07 | | | -0.01 (-1.37, 1.34) | 0.99 | | |

Secs: seconds; Mins: minutes; BW: Body weight; GRF: Ground reaction force

a Coefficients represent the change in y for a 10s change in duration of leg lifts

b Coefficients represent the change in y for a 10 repetitions of movement

c Coefficients represent the change in Y for a 10° increase in angle

*Positive signed coefficient indicated increase in the independent variable associated with increased pain severity

#Negatively signed coefficient indicated increase in the independent variable associated with increase in pain related disabilityWhen considering movement quantity, there was no evidence of between-person associations with pain severity after adjusting for year and stream.

Considering pain related disability, there was evidence for some weak between person associations for movement quantity. At a between-person level, greater levels of disability were associated with larger amounts of time spent in light intensity activity. Additionally, greater levels of disability were associated with a lower leg lift count to the front and overall, as well as less time spent performing leg lifts to the front. It is widely cited that dancers frequently continue to dance despite the presence of pain and related disability [36, 37]. The results of our research suggest that dancers continue to dance when experiencing pain with small modifications of movement quantity. It is possible that these findings reflect an adaptive response to reduce load, while continuing to dance with disabling pain. An alternative hypothesis is that the observed reduction in load is indicative of dancers' lack of strength, which may in turn lead to increased pain related disability. Further research involving daily dancer monitoring and temporal analysis would provide indication of causality and the bidirectional relationship of movement and pain.

There was also evidence for some weak associations between pain related disability and movement quality, specifically, greater pain related disability was associated with greater thigh elevation angles during side leg lifts. Initially this came as a surprise as movement is thought to be more constrained in the presence of pain and disability [38], however a systematic review has identified that when experiencing disabling pain, people move with greater movement

variability [39]. Considering these findings together, it could be hypothesised that dancers with higher levels of disability were modifying their training behaviours in some ways to reduce general load (e.g., reducing the volume of movement), while also modifying behaviours in ways which may increase specific joint loading (e.g., pushing how high they lifted their leg during side leg lifts). However, it cannot be assumed that all dancers were employing these strategies. Indeed, the low number of differences in movemen, and with no clear pattern, may suggest that dancers are generally able to maintain movement quality despite pain. Additionally, analysis did not account for the specific movements that dancers reported as provocative. Thus, moving forwards in unravelling the individual complexity of the relationship between movement quality and quantity and pain, it may be more helpful to evaluate these associations, using serial monitoring of individual dancers rather than investigating group differences.

Interestingly, there was no evidence for an association between either pain severity or pain related disability and any jumping variables, suggesting that the dancers may be able to maintain movement quantity and quality irrespective of pain, during jumping activity. This finding challenges findings of a previous cross-sectional study comparing dancers with and without anterior knee pain (n = 25) suggested that those with pain demonstrated greater peak GRF during a ballet specific jump (mean difference 0.2BW, CI: 0.08, 0.32) [14]. However, this study was limited in that it only captured a single trial of a jump on a single day, and only considered pain presence rather than pain severity. In contrast, we captured all jumps within the dancers' daily class at 4 different time points, taking into consideration changes in pain severity at each of these time points. Another explanation may relate to the relatively low levels of pain observed in this study, with average pain scores ranging from 3.1–3.5/10 across the 4 time points. It is possible that the threshold for marked changes in movement in response to pain was not met, in line with previous experimental research [40]. Finally, pain location may also influence these findings, where it can be hypothesised that a dancer with foot, ankle or knee pain may land differently to a dancer with hip or low back pain. With serial monitoring of individual dancers in future research, the relationship between movement quality variables specific pain locations and the changes in pain severity relative to these locations may be explored.

## Strengths and limitations

Previous research in this space has focussed predominantly on general movement quantity with limited focus on specific movement quantity and movement quality. Thus, a major strength of this study is the unique combination of movement quantity and quality measures and the exploration of their relationship with pain and disability. Importantly, this study used technology that allowed for field-based measurement of movement in a dancer's normal ballet class environment. This system was able to detect common ballet movements that are considered potentially pain provocative. The serial design allowed for observation at the within-person level, however the use of only 4 time points was a limitation as discussed below.

The study was limited to a sample of pre-professional female dancers from a single dance training facility. To promote generalisability, future research should include dancers from multiple centres and include male and non-binary dancers as variations in training regimes across facilities, and gender specific movement profiles, may influence results. For example, the greater volume of jumping activities performed by male dancers, could mean jumps are more strongly associated with pain and/ or disability in men.

The fact that we only identified evidence for between-person associations may be a result of the study design. Repeated measures at only 4 time points over a 12-week period rather than daily monitoring, reduced the information available to elucidate associations at the level of the

individual dancer. Furthermore, the variation in pain location, and task selection for the PSFS, over the 4 time points both within dancers across time, as well as between dancers, was not possible to account for within the analysis with the numbers available. The study also only considered physical factors associated with pain and disability, when pain is known to be a complicated biopsychosocial construct [41]. This study is proof-of-concept that the field-based system used could allow future research to evaluate these associations at an individual level using serial monitoring over time. This would provide adequate data for sophisticated temporal analyses to further unravel the complexities of dancers' pain and disability.

To our knowledge this is the first application of machine learning to wearable sensor data that has been used in a longitudinal field-based study. However, a number of challenges of the wearable sensor system need to be addressed before more sophisticated applications of the wearable sensor system can be undertaken. The system required multiple sensors being attached to each dancer and, as detailed in Fig 2, there were several steps in the processing of data due to multiple machine learning models which, combined with the computational demands of the machine learning algorithms, resulted in lengthy data processing. Additionally, while the human activity recognition model used demonstrated an acceptable degree of accuracy when validated in previous work [25], the application of this system in a true field-based study required extensive manual data cleaning as the accuracy of classification varied amongst the dancers. Specifically, each identified movement was visually inspected by 1 of the researchers and removed if the movement was misclassified. This was why only a single dance class was analysed on each of the 4 days. All previous applications have focused on the development and validation of systems, thus have not accounted the extensive data processing and cleaning that is required when these novel models are applied in field-based settings [42]. To allow for larger studies with continuous daily monitoring the machine learning models would require further optimisation, to allow for a fully automated accurate system.

## Clinical implications and conclusions

The results of this study provide insight into how dancers with disabling pain may adapt the way they move to reduce load, in order to continue dancing. However, it is unlikely that these same responses are adopted by all dancers when faced with pain and pain related disability. Further, it is likely that complex interactions between movement quantity and quality, as well as other biopsychosocial factors, that are unique to each individual, influence a person's pain development and coping responses to pain. Future application of wearable sensor technology provides the opportunity for clinicians to gain a deeper insight into the inter-relationships between pain, disability, and movement in athletic populations, to better inform person centred care.

The field-based sensor system used in this research can provide quantitative information on both movement quantity and quality in a real-world environment. While further optimisation of the technology used in this research is needed to promote ease of usability, this research demonstrates a proof-of-concept for larger, longitudinal field-based research to occur. Specifically, it provides future opportunity using frequent, field-based, serial measures of movement quantity and quality in a dancer's everyday training, to allow the collection of the large amount of data needed for modelling the complexity of interrelationships between movement, pain, disability and other salient factors, using sophisticated analytics such as complex systems approaches [43]. This creates opportunities within clinical research and practice for assessment and monitoring of individual dancers, and detect shifts in individual dancer movement behaviours in response to treatment or advice.

## Author Contributions

**Conceptualization:** Danica Hendry, Amity Campbell, Leon Straker, Peter O'Sullivan.

**Data curation:** Danica Hendry, Luke Hopper, Leon Straker, Peter O'Sullivan.

**Formal analysis:** Danica Hendry, Amity Campbell, Anne Smith, Leon Straker, Peter O'Sullivan.

**Investigation:** Danica Hendry, Leon Straker, Peter O'Sullivan.

**Methodology:** Danica Hendry, Amity Campbell, Anne Smith, Leon Straker, Peter O'Sullivan.

**Project administration:** Danica Hendry, Amity Campbell, Luke Hopper, Leon Straker.

**Supervision:** Amity Campbell, Leon Straker, Peter O'Sullivan.

**Writing – original draft:** Danica Hendry, Amity Campbell, Anne Smith, Leon Straker, Peter O'Sullivan.

**Writing – review & editing:** Danica Hendry, Amity Campbell, Anne Smith, Leon Straker, Peter O'Sullivan.

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
