## [Decision Letter · Decision Letter 0]

13 Dec 2021

PONE-D-21-29319Movement quantity and quality: How do they relate to pain and disability in dancers?PLOS ONE

Dear Dr. Hendry,

Thank you for submitting your manuscript to PLOS ONE. After careful consideration, we feel that it has merit but does not fully meet PLOS ONE’s publication criteria as it currently stands. Therefore, we invite you to submit a revised version of the manuscript that addresses the points raised during the review process.

We look forward to receiving your revised manuscript.

Kind regards,

Nili Steinberg

Academic Editor

PLOS ONE

Journal Requirements:

DH is the recipient of an Australian Research Training Program Scholarship (RTP) (no grant number) for her PhD research of which this study was part of. 

Additional Editor Comments:

Please notice that the reviewers suggested that not all the data underlying the findings in your manuscript is fully available

Reviewers' comments:

Reviewer's Responses to Questions

**Comments to the Author**

1. Is the manuscript technically sound, and do the data support the conclusions?

Reviewer #1: Yes

Reviewer #2: Yes

2. Has the statistical analysis been performed appropriately and rigorously? 

Reviewer #1: Yes

Reviewer #2: Yes

3. Have the authors made all data underlying the findings in their manuscript fully available?

Reviewer #1: Yes

Reviewer #2: No

4. Is the manuscript presented in an intelligible fashion and written in standard English?

Reviewer #1: Yes

Reviewer #2: Yes

5. Review Comments to the Author

Reviewer #1: This is a well written and well-designed study. It examines an important topic – dance/ exercise and pain – and uses novel technology to provide new insight into the relationship between movement and pain. The study has a relatively large and representative sample considering the specialist population and methods. It features valid assessment of the exposure – pain – and data collected at multiple time points. There are several audiences that may be interested in these findings – dance/ exercise/ movement scientists, rehabilitation professionals, teachers/ coaches, and engineers. This paper should be considered for publication.

My primary concern – as detailed below – is the number of significant findings vs the total number of outcomes examined. The very low number of differences does not support a clear relationship between pain and movement execution in dancers. Yet the reporting and discussion seems locked in on these few differences they do not consider the alternative finding: that pain, despite its prevalence in dance, has limited discernible impact on movement quality.

Comments:

Methods

Lines 135 – movement quality. I do not believe the authors have provided a clear definition of movement quality or quantity. Does quality refer to range of motion? Or does it incorporate other elements. Could the authors please provide a lay description – e.g., ‘movement quality was defined as …’ – that a physical therapist/ movement coach could appreciate without additional background reading.

Lines 136. Are these inertial sensors? Please say more than the specific model. The authors should also say the type of sensor (e.g., accelerometer, inclinometer, inertial measurement unit) and – ideally – for more novel sensors provide a definition of sensor type.

Lines 137 – 138. I do not mind stating ‘previously detailed’ – but think it is important to provide some brief information. Few readers will want to troll back through old papers - so stating the sites – e.g., hip, low back, etc – would be of value.

Lines 165 - 175: I'll put this down to my ignorance - but I am not sure how you get between group differences for an intensity score between 0 - 10? When I think of 'between group', I think of has pain vs does not have pain OR has no pain, has some pain, has high pain. I trust the authors have this right but perhaps they can clarify the terminology used?

Results

Lines 216 – 217: “We identified a few modest associations between dancers’ movement quantity and quality and dancers’ self-reported pain outcomes.”

The authors looked at 10 possible between group differences for pain severity/ disability and general movement quality. There was only one significant finding after adjustment. They looked at – by my count – 42 between group differences for specific movement quality parameters. Only four were significant. There do not seem to be any within group differences.

The results should state more explicitly that of the large number of movement parameters examined, only a very small number of differences were identified. It is OK to say pain did not appear to have a large impact on movement in this sample.

Lines 226 – 227: “When considering movement quantity, there was some evidence of between-person associations with pain related disability on adjusted analysis”.

Again, there may have been some significant differences, but on the whole these were minimal. I recommend the authors tone change from emphasising that there was some difference to stating that on balance differences were minor.

Lines 227 – 229. “A 1-minute increase in light activity was associated with an increase in pain related disability of 0.15 points (95%CI: -0.26, -0.03, p=0.02).” Is this correct or should the effect estimate be -0.15? Otherwise it is outside the CI.

Discussion

Not of consequence but the line numbers disappear here.

Paragraph 1: “While there was no evidence of associations with dancers self-reported pain severity, associations were identified between some movement factors and pain related disability.”

Again, please consider qualifying this statement. The authors examined a lot of angles. <10% of the angles they examined were different.

Page 20 paragraph 1. “The results of our research suggest that while dancers continue to dance when experiencing pain, they do so with modifications. It is possible that these findings reflect an adaptive response to reduce load, while continuing to dance with disabling pain.”

The results may suggest that dancers make modifications. But there were not many changes and there was no clear pattern. The very low number of differences and the very high number of outcomes examined may also suggest that dancers are able to maintain movement quality despite pain. I feel there is a reluctance to acknowledge this and an eagerness to focus on the few angles that were different. I am not arguing the authors need to shy away from what they did identify – but I think it is important to acknowledge that perspective. Sorry to bemoan this point - it is a good paper.

Reviewer #2: Thank you for the opportunity to review this interesting study that examines potential relationships between dancers’ movement quantity and quality and their experience of pain and pain related disability. The study was ambitious, involving 52 pre-professional dance students who were assessed at four time points across 12 weeks, using both self-report measures and wearable movement sensors. The results indicate that pain is common among dancers and disabling in half of dancers. Particular movements were associated with a greater likelihood of pain related disability. These results provide guidance to dancers and dance organisations about how to prevent movement related pain and disability and to sustain dancers’ careers in good health. They also provide methods that can be successfully used in field research on dance. I make the following suggestions for how the paper could be further developed and clarified.

The abstract could mention that these were ballet and contemporary dance students (as opposed to other forms of dance).

Introduction p. 4, GRF – provide full wording the first time the abbreviation is used.

Methods – the measures were clearly described although there are a number of single item measures. Is there a reference to the validity of such measures? The Figures with the assessment schedule and the flow chart of the study procedure are helpful.

Results – linear mixed models were used. Was the level 1 variable time (p. 8, line 165)?

Discussion- the conclusion on p. 19 that most dancers continued dancing despite being in pain, with some training modifications ‘which may reflect relatively low levels of pain and disability’ is interesting. An alternative conclusion is that in the context of competitive auditions for dance corps, there may be a culture of denial and under-reporting of pain in dancers which is to the detriment of their recovery and career longevity. Perhaps a more balanced conclusion could be presented here?

In the limitations section where the authors correctly mention that the study cannot be generalised to male dancers (or non-binary participants, to be inclusive), could the findings in relation to types of movements and pain disability risk point to any potential gender effects in future research? That is, are some types of dance movements more common in female vs non-female dancers and if so, what is their pain experience likely to be?

The paragraphing needs work in places, e.g., the last sentence on lines 178-179 should be joined with the previous sentence to make a more balanced paragraph. Similarly, there is a single sentence paragraph on p. 12. The images in figures 1 and 3 are blurry in the pdf document.

6. PLOS authors have the option to publish the peer review history of their article (what does this mean?). If published, this will include your full peer review and any attached files.

Reviewer #1: **Yes: **C Swain

Reviewer #2: No

---

## [Author Response · Author response to Decision Letter 0]

22 Mar 2022

We thank the reviewer for their comments. Please find the response to reviewers within the submission

---

## [Decision Letter · Decision Letter 1]

1 May 2022

Movement quantity and quality: How do they relate to pain and disability in dancers?

PONE-D-21-29319R1

Dear Dr. Hendry,

We’re pleased to inform you that your manuscript has been judged scientifically suitable for publication and will be formally accepted for publication once it meets all outstanding technical requirements.

Kind regards,

Nili Steinberg

Academic Editor

PLOS ONE

Additional Editor Comments (optional):

Reviewers' comments:

Reviewer's Responses to Questions

**Comments to the Author**

1. If the authors have adequately addressed your comments raised in a previous round of review and you feel that this manuscript is now acceptable for publication, you may indicate that here to bypass the “Comments to the Author” section, enter your conflict of interest statement in the “Confidential to Editor” section, and submit your "Accept" recommendation.

Reviewer #1: (No Response)

Reviewer #2: All comments have been addressed

2. Is the manuscript technically sound, and do the data support the conclusions?

Reviewer #1: Yes

Reviewer #2: Yes

3. Has the statistical analysis been performed appropriately and rigorously? 

Reviewer #1: Yes

Reviewer #2: Yes

4. Have the authors made all data underlying the findings in their manuscript fully available?

Reviewer #1: No

Reviewer #2: No

5. Is the manuscript presented in an intelligible fashion and written in standard English?

Reviewer #1: Yes

Reviewer #2: Yes

6. Review Comments to the Author

Reviewer #1: (No Response)

Reviewer #2: The authors have responded thoughtfully to the issues raised by me and the other reviewer. Their paper is reading very well now

7. PLOS authors have the option to publish the peer review history of their article (what does this mean?). If published, this will include your full peer review and any attached files.

Reviewer #1: No

Reviewer #2: **Yes: **Genevieve Dingle

---

## [Editor Report · Acceptance letter]

10 May 2022

PONE-D-21-29319R1 

Movement quantity and quality: How do they relate to pain and disability in dancers? 

Dear Dr. Hendry:

I'm pleased to inform you that your manuscript has been deemed suitable for publication in PLOS ONE. Congratulations! Your manuscript is now with our production department. 

Kind regards, 

on behalf of

Prof. Nili Steinberg 

Academic Editor

PLOS ONE